# Spatial and Temporal Distribution and the Driving Factors of Carbon Emissions from Urban Production Energy Consumption

**DOI:** 10.3390/ijerph191912441

**Published:** 2022-09-29

**Authors:** Liyuan Fu, Qing Wang

**Affiliations:** School of Economics, Liaoning University, Shenyang 110036, China

**Keywords:** carbon emissions from production energy consumption, regional imbalance, carbon emission classification, driving factor

## Abstract

Urban production energy consumption produces a large amount of carbon emissions, which is an important source of global warming. This study measures the quantity and intensity of carbon emissions in 30 provinces of China based on urban production energy consumption from 2005–2019, and uses the Dagum Gini coefficient, kernel density estimation, carbon emission classification and spatial econometric model to analyze the spatial and temporal distribution and driving factors of quantity and intensity of carbon emissions from China and regional production energy consumption. It was found that the growth rate of carbon emission quantity and carbon emission intensity of production energy consumption decreased year by year in each province during the study period. The imbalance of carbon emission was strong, with different degrees of increase and decrease, and there were big differences between eastern and western regions. The classification of carbon emissions differed among provinces and there was heterogeneity among regions. The quantity and intensity of carbon emissions of production energy consumption qwre affected by multiple factors, such as industrial structure. This study provides an in-depth comparison of the spatial and temporal distribution and driving factors of quantity and intensity of carbon emissions of production energy consumption across the country and regions, and provides targeted policies for carbon emission reduction across the country and regions, so as to help achieve China’s “double carbon” target quickly and effectively.

## 1. Introduction

The massive emission of greenhouse gases has led to an increase in global temperature and an accelerated frequency of extreme hot weather, which has seriously damaged ecological health [1], and has also affected the productive life of human beings [2]. The problem of climate warming is no longer just a problem for individual countries, but a problem for all mankind [3]. The impact of warming caused by carbon emissions is stronger in China, where the average temperature has risen by 0.26 degrees per decade from 1961–2020 in China, which is higher than the global average. China’s environmental problems, such as glacier melting, permafrost degradation and natural disasters, have intensified in recent years. The largest mountain glaciers in the northeast of the Qinghai-Tibet Plateau have dropped 450 m in the last 70 years, and the perennial permafrost in northeast China has degraded significantly, with the southern boundary of permafrost degradation exhibiting phenomena such as northward shift and expansion of the ablation zone. The maximum seasonal freezing depth reduction rate in Heilongjiang Province, China, was about 9.9 cm per decade from 1961–2016. Permafrost degradation can lead to significant impacts on regional ecology, hydrological processes, carbon cycle, and cold zone engineering construction and operation. In addition, extreme and severe weather in China, such as heavy rainfall, floods and mountain fires, seriously affect human activities and endanger human life and health. To address the global climate change issue, the United Nations Intergovernmental Panel on Climate Change (IPCC) stated that measures should be taken to limit the global temperature increase to 1.5 degrees [4]. In its sixth climate report, the IPCC stated that in order to control global warming, greenhouse gas emissions need to peak in 2025 and be reduced by 43% from 2010 levels by 2030, so deep cuts in emissions are needed across all sectors. In response to global climate change, China has committed to achieve carbon peaking by 2030 and carbon neutrality by 2060 [5], which means that China needs to ensure sustainable economic and social development and energy security, while developing technologies and implementing policies to quickly and effectively curb carbon emissions and achieve carbon peaking in a short period of time. According to the carbon emission data of the World Bank, China’s carbon emissions increased at an average annual rate of 5.98% from 1996 to 2018 [6]. As the world’s largest developing country and carbon emitter [7], the task of achieving carbon peaking and carbon neutrality in China is still very difficult. Currently, the global energy sector produces about 25% of global carbon emissions, and China’s energy use is still dominated by fossil energy sources such as coal, which accounts for nearly 56.8% of the energy mix and produces a high amount of carbon emissions and is highly polluting to the atmosphere [8]. China has begun to actively pursue low-carbon development, optimize its energy consumption structure, and promote high-quality economic and social development [9,10].

## 2. Literature Review

As a region where human activities are concentrated, China has seen a significant increase in carbon emissions due to the production of various industries in the process of urbanization and industrialization, and the contradiction between economic development and resource consumption and environmental pollution problems has become more and more prominent [11,12,13,14,15]. Some provinces in China have air pollution levels that are among the highest in the world [16], due to rapid economic growth, a large increase in the number of motor vehicles and a large population leading to increasingly serious air pollution [17,18]. Therefore, in recent years, China has made great efforts to develop a clean coal power system in terms of energy consumption and introduced several policies to promote the clean and efficient use of coal [19], but China’s energy sources, such as oil and natural gas, are scarce, and the dependence on coal in various industries remains high [20]. The current per capita energy consumption of China is relatively low compared with the international level, and there are also imbalances in development between provinces within China. Imbalanced per capita energy consumption levels often lead to imbalanced economic development, and this also means inefficient use of energy, which poses a major challenge to China’s energy security and carbon emission reduction goals [21,22]. In order to seek a better path for reducing carbon emissions on the basis of energy security, while ensuring smooth economic operation, scholars have conducted a lot of research on energy and carbon emission related areas [23,24,25].

There are many factors affecting carbon emissions, mainly in economic development, energy use, population structure, urban development, etc. Relevant studies show that China’s carbon emissions have a strong correlation with the level of economic development, and China’s industrial carbon emissions show an inverted U-shaped non-linear relationship with the level of economic development [26]. Measuring the relationship between economic growth rate and carbon emission from the perspective of economic growth rate, it is found that the development of economic growth rate and carbon emission growth rate is consistent, and has the characteristics of stages [27]. Carbon emissions are also usually lower than normal at economically backward stages, and the level of urbanization shows an inverted U-shaped correlation with carbon emissions when energy resources are not fully exploited [28]. Xu et al. (2016) argue that the combined effects of economic output and energy efficiency play an important role in carbon emissions at different stages, while energy structure has a smaller effect on carbon emissions [29]. Li et al. (2018) found that economic development has a relatively limited effect on carbon emissions, and that energy consumption structure is an important cause of higher carbon emission levels. Meanwhile, the effects of industrial structure and energy efficiency on carbon emissions in different regions are heterogeneous [30]. The analysis of carbon emission driving factors at different stages in Beijing found that different factors have different degrees of influence on carbon emissions at different stages of urban development. Economic output has always been the most important contributor to carbon emissions in Beijing at different stages, and population size and energy structure also have a promoting effect on carbon emissions [31]. Urbanization effect, resident consumption effect and population size effect have significant promoting effects on carbon emissions, and population urbanization is the most important population influencing factor in increasing carbon emissions [32].

The influence of different indicators on carbon emissions shows the diversity of carbon emission pathways, and in the process of economic and social development, large consumption of energy is one of the important sources of carbon emissions. Existing study shows that energy consumption and carbon emissions show a certain spatial correlation [33], and the different industrial structures and economic growth modes have heterogeneous dependence on energy, which leads to a diversity of carbon related emissions [34]. The energy intensity of production sectors is the main driving factor of carbon emission intensity [35], and has a positive impact on carbon emission intensity [36]. The production activities of each industry are closely linked to carbon emissions, and the carbon emissions of China’s construction industry show the characteristics of network structure, and the geographical proximity and energy intensity are all significantly correlated with the carbon emissions of the construction industry [37]. China’s manufacturing carbon emissions are influenced by industrial value added, energy consumption and energy structure, and it is found that the decoupling of manufacturing carbon emissions from economic development is mainly dominated by the decoupling of energy consumption from industrial value added [38], which further shows the importance of energy consumption on carbon emissions. A survey of carbon emissions in China’s internal regions and different provinces and cities found that carbon emissions from urban energy consumption in 26 prefecture-level cities in the Yangtze River Delta will be stable by 2020, and carbon emission intensity will decline, with carbon emissions from energy consumption and economic development showing a negative decoupling trend [39]. Zhang et al. (2021) used the input–output model to measure the difference of carbon emissions due to production-based and consumption-based activities in Tianjin, China, and found that basic building construction was the main reason for the difference, and found that population, income and urbanization had an important impact on urban carbon emissions [40]. Ma et al. (2022) explored the influencing factors of carbon emissions from the energy consumption of rural residents in China and found that quality of life, energy conservation awareness and household characteristics would affect the carbon emission level of rural residents [41].

In addition, scholars have conducted a lot of research on the regional nature of carbon emissions. Chen et al. (2022) used the Theil index and the Moran index to conduct spatial analysis of China’s carbon emission intensity and found that the spatial differences of carbon emission intensity from 2000 to 2019 were obvious, mainly intra-regional, with significant spatial correlation and local agglomeration characteristics [42]. Yang et al. (2022) analyzed the spatial and temporal patterns of carbon emission quantities in China’s prefecture-level cities through spatial autocorrelation models and found that China’s prefecture-level carbon emission quantities expanded in time, decreased from north to south, and increased from southeast to northwest, with significant spatial aggregation [43]. Liu et al. (2022) found that the carbon emission reduction effect is significant in northeast, east and southwest China, while the carbon emission reduction effect is poor in northwest China, and investment activities, energy use and economic activities promote carbon emissions in China [44]. Wang and Zhao (2021) established an industrial carbon emission performance (ICEP) evaluation system to study the regional industrial carbon emission levels and found that industrial carbon emission performance has significant inter-provincial differences, and, by region, the ICEP was highest in the eastern region and lowest in the western region [45]. Zhang and Li (2022) measured and analyzed the carbon emissions from energy consumption of rural residents and agricultural production in China and found that rural carbon emissions in China’s provinces have spatial agglomeration, and carbon emissions in the eight economic regions have large inter-regional differences and small intra-regional differences [46]. Xiao et al. (2022) analyzed the level and spatio-temporal characteristics of county carbon emissions in Hubei Province from 2000 to 2020 and found that carbon emissions in central and eastern areas were higher than those in western. Moreover, carbon emissions in Hubei Province were significantly decoupled from agricultural economic growth, and the number of counties with strong decoupling has increased [47].

In general, the existing studies on carbon emissions are rich, but there are still certain shortcomings. Firstly, existing studies have deeply analyzed the relevant influencing factors of carbon emissions, which can clearly sort out the theoretical relationship between carbon emissions and related influencing factors. However, carbon emissions have a spatial nature, and current studies lack discussion on the spatial relationship between carbon emissions and their influencing factors. Secondly, carbon emissions in China come from a wide range of sources, and carbon emissions from energy consumption are a key source in the process of industrialization and urbanization in China, so research on carbon emissions from energy consumption is necessary, and most existing studies have been conducted on carbon emissions from energy consumption. Energy consumption can be divided into rural energy consumption and urban energy consumption, and the corresponding urban energy consumption is divided into production energy consumption and domestic energy consumption. A more detailed division of energy consumption sources and the study of its carbon emission nature and influencing factors can propose more targeted energy saving and emission reduction policies. Thirdly, carbon emissions are spatially imbalanced, but the degree of imbalance and how it evolves still need to be measured and analyzed further in order to clarify the regional differences and spatial distribution characteristics of carbon emissions. Based on the above three analyses, this paper decided to measure the carbon emission quantity and carbon emission intensity of urban production energy consumption, and analyze the spatial and temporal distribution of carbon emissions and the driving factors. The carbon emissions of urban production energy consumption refer to the carbon emissions generated by the energy consumption of production activities in industries, raw materials and materials, construction, transportation, storage, postal services, wholesale and retail trade, and accommodation and catering, in addition to agriculture. After measuring carbon emission quantity and carbon emission intensity, the national and regional imbalance and spatial distribution characteristics of carbon emissions from urban production energy consumption were further measured and analyzed using the Dagum Gini coefficient and its decomposition and kernel density estimation, and, finally, the driving factors of carbon emissions and the regional heterogeneity of the driving factors were investigated using spatial econometric models. The results of this paper aim to provide quantitative support for the formulation of national and regional carbon emission reduction policies, so as to achieve China’s “double carbon” target quickly and effectively.

## 3. Materials and Methods

### 3.1. Carbon Emission Calculation

In this paper, the carbon emission quantity and carbon emission intensity of urban production energy consumption were used to measure the regional carbon emission level, and the carbon emission quantity of urban production energy consumption was calculated using energy in industry, raw materials and materials, construction, transportation, storage, postal industry, wholesale, retail trade, accommodation, catering production and other end consumption.

#### 3.1.1. Carbon Emission Quantity Calculation

At present, there are many accounting methods for carbon emissions from energy consumption, including the IPCC measurement method, field measurement method and model estimation method [48]. In this paper, based on the urban production energy consumption and the method of calculating the carbon emission quantity of energy consumption according to IPCC [49], a total of 16 energy sources, including raw coal, washed coal, other washed coal, coal, coke, coking coal furnace gas, other coking products, crude oil, kerosene, diesel, fuel oil, liquefied petroleum gas, other petroleum products, natural gas, electricity and heat, were selected to calculate the carbon emissions from urban production energy consumption of 30 provinces in China from 2005 to 2019. The energy data were derived from the terminal consumption of the energy balance table in the China Energy Statistical Yearbook [50]. The calculation formula of carbon emission quantity is as follows:(1)CEQ=∑iEi×NCVi×EFi×Oi×4412

In Formula (1), *E_i_* is the energy consumption, *NCV_i_* is the average low level calorific value of the fuel, *EF_i_* is the carbon content per unit calorific value of the fuel, *O_i_* is the oxygen content, and 44/12 is the conversion factor for converting *C* to *CO_2_*.

#### 3.1.2. Carbon Emission Intensity Calculation

The carbon emission intensity of urban production energy consumption is the carbon emission of urban production energy consumption per unit of GDP. The calculation formula of carbon emission intensity is as follows:(2)$$CEI=\frac{Carbon\;emissions\;quantity\;of\;urban\;production\;energy\;consumption}{Value\;added\;of\;the\;secondary\;industry+Value\;added\;of\;the\;third\;industry}$$

In Formula (2), secondary industry refers to the mining industry (excluding mining auxiliary activities), manufacturing (including metal products, machinery and equipment repair), electric power, heat, gas and water production and supply industry, and construction industry. The third industry, or tertiary industry, refers to the service industry, including wholesale and retail, transportation, warehousing and postal services, accommodation and catering industries, etc.

### 3.2. Dagum Gini Coefficient

The Gini coefficient and its decomposition are important methods to study regional differences, and the sample is divided into three parts: between-group differences, within-group differences and hyper-variance density to explore regional differences and their sources in depth [51,52]. In this paper, the Gini coefficient and its decomposition were used to measure the regional differences in carbon emission levels of urban production energy consumption. The overall Gini coefficient reflected the overall differences in carbon emission levels of inter-provincial urban production energy consumption in China, and the specific formula is as follows:(3)$$G=\frac{\sum_{i=1}^{k}\sum_{m=1}^{k}\sum_{j=1}^{n_{i}}\sum_{r=1}^{n_{m}}\left|y_{ij}-y_{mr}\right|}{2n^{2}\mu }$$

The value *k* is the number of research subjects grouped. In this paper, it was divided into three regions: eastern, central and western China, containing a total of *n* research subjects. The values *y_ij_* and *y_mr_* are the carbon emission levels of the *j*(*r*)th province in the *i*(*m*)th research region, and *μ* is the average value of carbon emissions of all provinces.

The intra-regional Gini coefficient measures the difference in carbon emission levels within each region, and the inter-regional Gini coefficient measures the difference in carbon emission levels between regions. The specific formulae are as follows:(4)$$G_{ii}=\frac{\sum_{j=1}^{n_{i}}\sum_{r=1}^{n_{i}}\left|y_{ij}-y_{ir}\right|}{2n_{i}^{2}\mu_{i}}$$
(5)$$G_{im}=\frac{\sum_{j=1}^{n_{i}}\sum_{r=1}^{n_{m}}\left|y_{ij}-y_{mr}\right|}{n_{i}n_{m}\left(\mu_{i}+\mu_{m}\right)}\;$$
(6)$$\mu_{m}\leq \cdots \leq \mu_{i}\leq \cdots \leq \mu_{k}\;$$

*G_ii_* is the intra-regional Gini coefficient, *G_im_* is the inter-regional Gini coefficient, *n_i_* (*m*) and *μ_i_* (*m*) are the number of research objects in region *i* (*m*) and the mean carbon emission level of provinces in region *i* (*m*).

The overall Gini coefficient is decomposed to further measure the contribution of intra-regional differences, inter-regional differences and supervariable density to the overall differences. The formulae are as follows:(7)$$G=G_{w}+G_{nb}+G_{l},G_{w}=\sum_{i=1}^{k}G_{ii}p_{i}s_{i}$$
(8)$$G_{nb}=\sum_{i=2}^{k}\sum_{m=1}^{i-1}G_{im}\left(p_{i}s_{m}+p_{m}s_{i}\right)D_{im}$$
(9)$$G_{l}=\sum_{i=2}^{k}\sum_{m=1}^{i-1}G_{im}\left(p_{i}s_{m}+p_{m}s_{i}\right)(1-D_{im})$$
(10)$$D_{im}=\frac{d_{im}-p_{im}}{d_{im}+p_{im}}$$

*G_w_*, *G_nb_* and *G_l_* are the intra-regional, inter-regional and hypervariable density difference contributions, respectively. *D_im_* is the relative impact of carbon emission levels between regions *i* and *m*, *d_im_* is the difference in carbon emission levels between regions *i* and *m*, i.e., the mathematical expectation of the sum of all sample values of *y_ij_ − y_mr_* > 0 in regions *i* and *m*, and *p_im_* is the hypervariable first-order matrix, i.e., the mathematical expectation of the sum of all sample values of *y_ij_ − y_mr_* > 0 in regions *i* and *m*.

### 3.3. Kernel Density Estimation

Kernel density estimation is an important nonparametric estimation method, which can further explore the absolute differences and dynamic evolution of regional carbon emissions [53,54]. In this paper, we used the Gaussian kernel function for kernel density estimation to study the dynamic distribution and evolution of carbon emission levels of production energy consumption in inter-provincial cities in China from 2005 to 2019, and to analyze the agglomeration and dispersion of carbon emissions through the height and width of the wave. The formula is as follows:(11)$$f\left(x\right)=\frac{1}{ph}\sum_{i=1}^{p}K\left(\frac{y_{i}-\bar{y}}{h}\right)$$

*y_i_* is the carbon emission level of province *i*, $\bar{y}$ is the mean of carbon emission level of all provinces, *p* is the sample size, *h* is the bandwidth, and *K* is the Gaussian kernel function.

### 3.4. Carbon Emission Classification

To explore the changes of carbon emission levels of urban production energy consumption and their carbon emission types from 2005 to 2019, the carbon emissions of urban production energy consumption were divided into four categories, namely high-high (high carbon emission quantity-high carbon emission intensity), high-low (high carbon emission quantity-low carbon emission intensity), low-high (low carbon emission quantity-high carbon emission intensity) and low-low (low carbon emission quantity-low carbon emission intensity), using the average value of carbon emission quantity and carbon emission intensity as the measure. Those above the average value were high carbon emissions, and those below the average value of carbon emissions were low carbon emissions.

### 3.5. Model Setting

It is considered that the carbon emission level of urban energy production consumption in provincial in China may have the characteristics of spatial correlation. This paper proposed to test the drivers of carbon emission levels of production energy consumption in towns using the spatial error model (SEM), spatial lag model (SLM) and spatial Durbin model (SDM). The spatial correlation of carbon emission levels of urban production energy consumption in China was tested using the global Moran index (Global Moran’s I) [55]. If the test was passed, a spatial panel model was required, and then LM, Wald and LR tests [56] were conducted to select the three spatial models SEM, SLM and SDM, and the fixed and random effects were selected by the Huasman test.

(1)Spatial error model (SEM)


(12)
$$\begin{array}{l} \ln CE_{it}=\beta_{1}\ln IS_{it}+\beta_{2}\ln FDI_{it}+\beta_{3}\ln PCDI_{it}+\beta_{4}\ln ECS_{it}+\beta_{5}\ln DREP_{it}\\ +\beta_{6}\ln PD_{it}+\beta_{7}\ln GCR_{it}+\beta_{8}\ln UR_{it}+\beta_{9}\ln CL_{it}+\beta_{10}\ln STI_{it}+\mu_{i}+\eta_{t}+u_{it} \end{array}$$



(13)
$$u_{it}=\lambda \sum_{j=1}^{n}w_{ij}u_{jt}+\varepsilon_{it}\;$$


(2)Spatial lag model (SLM)


(14)
$$\begin{array}{l} \ln CE_{it}=\rho \sum_{j=1}^{n}w_{ij}\ln CE_{jt}+\beta_{1}\ln IS_{it}+\beta_{2}\ln FDI_{it}+\beta_{3}\ln PCDI_{it}+\beta_{4}\ln ECS_{it}\\ +\beta_{5}\ln DREP_{it}+\beta_{6}\ln PD_{it}+\beta_{7}\ln GCR_{it}+\beta_{8}\ln UR_{it}+\beta_{9}\ln CL_{it}+\beta_{10}\ln STI_{it}\\ +\mu_{i}+\eta_{t}+\varepsilon_{it} \end{array}$$


(3)Spatial Durbin Model (SDM)


(15)
$$\begin{array}{l} \ln CE_{it}=\rho \sum_{j=1}^{n}w_{ij}\ln CE_{jt}+\beta_{1}\ln IS_{it}+\beta_{2}\ln FDI_{it}+\beta_{3}\ln PCDI_{it}+\beta_{4}\ln ECS_{it}\\ +\beta_{5}\ln DREP_{it}+\beta_{6}\ln PD_{it}+\beta_{7}\ln GCR_{it}+\beta_{8}\ln UR_{it}+\beta_{9}\ln CL_{it}+\beta_{10}\ln STI_{it}\\ +\mu_{i}+\eta_{t}+\gamma \sum_{j=1}^{n}w_{ij}x_{jt}+\varepsilon_{it} \end{array}$$


In the formulae, *i*, *j* represents a province, t represents the year, *CE_it_* is the carbon emission level of the *i*th province in year *t*, *β* is the coefficient of the explanatory variable, *μ_i_* is the individual fixed effect, *η_t_* is the time fixed effect, *w* is the spatial weight matrix, *λ* is the spatial error term coefficient, *ρ* is the spatial autocorrelation coefficient, *x* is the explanatory variable, and *γ* is the coefficient of the explanatory variable of the neighboring province affecting the home province.

In this paper, the spatial weight matrix used the adjacency space weight matrix, and the neighborhood under the spatial structure was represented by 0 and 1. If the spatial units had a non-zero common boundary, they were considered to be spatially adjacent and represented by 1; otherwise, they were considered to be non-spatially adjacent and represented by 0.
(16)$$w_{ij}=\{\begin{array}{c} 1,\;\\ 0,\; \end{array}\begin{array}{c} \mathrm{region}\;i\;\mathrm{and}\;\mathrm{region}\;j\;\mathrm{are}\;\mathrm{adjacent}\\ \mathrm{others} \end{array}$$

### 3.6. Explanatory Variables Selection and Description

The driving factors of carbon emissions from urban production energy consumption are complex. In this paper, 11 explanatory variables were selected for analysis from five dimensions as the driving factors of carbon emissions from urban production energy consumption: urban economic level, living standard of urban residents, urban energy consumption level, urban population size and urban development level.

(1)Urban economic level. The transformation of industrial structure can be realized through the upgrading of industrial structure and the rationalization of industrial structure to reduce carbon emissions, so industrial structure is an important influencing factor of carbon emissions [57,58]. Foreign direct investment (FDI) has a significant spatial correlation with carbon emissions, and FDI has a significant impact on the carbon emission intensity of local and surrounding areas [59]. Therefore, the industrial structure (IS) and foreign direct investment (FDI) were selected to reflect the urban economic level. The proportion of value added in the secondary industry to GDP was used to measure the industrial structure.(2)Living standard of urban residents. Wen and Zhang found that per capita disposable income has a significant impact on carbon emissions [60]. Therefore, the per capita disposable income (PCDI) and per capita consumption expenditure (PCCE) in urban areas were chosen to reflect the living standard of urban residents, and, among them, per capita consumption expenditure replaced per capita disposable income for test robustness.(3)Urban energy consumption level. Studies have shown that the high proportion of coal consumption in China directly determines the energy consumption structure, which, in turn, is the driving factor of carbon emissions [61]. Therefore, the energy consumption structure (ECS) was chosen to reflect the urban energy consumption level. Coal is the main source of CO_2_ emissions, and the proportion of coal consumption to total energy consumption was used to measure the energy consumption structure.(4)Urban population size. As the main body of economic development, the population structure has a profound impact on carbon emissions. Labor force and dependency ratio are important demographic indicators, and have significant space differences in the impact of carbon emissions [62]. In addition, studies have shown that the population density of contribution to carbon emissions is high in the short-term and long-term, and population density is a non-negligible factor affecting carbon emission [63,64]. Therefore, the dependency ratio of elderly population (DREP) and population density (PD) were selected to reflect urban population size.(5)Urban development level. Studies have shown that the green coverage of built-up areas has a significant impact on provincial carbon emissions in China [65]. The spatial imbalance of per capita carbon dioxide level in China is obvious, and the urbanization rate is an important driving factor of carbon emissions [66]. Through the study of BRICS countries (Brazil, India, China, etc.), it was found that education level has a significant effect on carbon emissions, which can play a role in environmental quality [67], and scientific and technological innovation can affect carbon emissions by improving the energy intensity of high-tech industries [68]. Therefore, the green coverage rate of built-up area (GCR), urbanization rate (UR), cultural level (CL) and scientific and technological innovation (STI) were selected to reflect urban development level.

The urbanization rate was measured by the proportion of urban population in the total population, and the education level was measured by the average years of education: number of primary school students *6+ number of junior middle school students *3+ number of senior high school students *3+ number of junior college students and above *16/number of people aged 6 and above. Technological innovation was measured by the turnover of the technology market. The original data of all the explanatory variables were obtained from China Statistical Yearbook and China Energy Statistical Yearbook from 2005–2019, and individual missing data were completed using the trend prediction method and interpolation method. The abbreviations, definitions, and data sources for variables are shown in Table 1 and descriptive statistics are shown in Table 2.

### 3.7. Research Area

The carbon emission level of production energy consumption varies widely among different regions, and this paper divided China into three study regions: the eastern region, including Beijing, Tianjin, Hebei, Liaoning, Shanghai, Jiangsu, Zhejiang, Fujian, Shandong, Guangdong, and Hainan; the central region, including Shanxi, Jilin, Heilongjiang, Anhui, Jiangxi, Henan, Hubei, and Hunan; the western region, including Inner Mongolia, Guangxi, Chongqing, Sichuan, Guizhou, Yunnan, Shaanxi, Gansu, Qinghai, Ningxia, Xinjiang. The study area is shown in Figure 1.

## 4. Results and Discussion

### 4.1. Spatial and Temporal Distribution of Carbon Emissions from Urban Production Energy Consumption

#### 4.1.1. Spatial Distribution of Carbon Emissions

The results of calculating carbon emissions of urban production energy consumption in inter-provincial in China from 2005 to 2019 showed that carbon emission levels have obvious spatial distribution characteristics (Figure 2). Carbon emission quantity showed the regional characteristics of eastern region > central region > western region in 2005, while carbon emission intensity was opposite to the distribution of carbon emission quantity, and the western region showed more obvious differences to other regions, and its carbon emission intensity was much higher than that of other regions. By 2019, the spatial distribution of carbon emission quantity and carbon emission intensity had changed to a certain extent. The carbon emission quantity still followed the regional characteristics of eastern region > central region > western region, while the regional characteristics of carbon emission intensity of western region > central region > eastern region were more obvious than in 2005. The change in pattern of carbon emission levels from 2005 to 2019 showed the change of urban production energy consumption, the rough development mode leading to a continuous rise in total energy consumption, which, in turn, led to the rise of carbon emission quantity, but the annual growth rate of carbon emission quantity from 2005 to 2019 showed a decreasing trend. The carbon emission intensity from 2005 to 2019 also had an obvious decreasing trend. The inter-provincial average carbon emission intensity decreased from 3.7673 to 1.5340, a decrease of 59.28%, while the regional carbon emission intensity declined, showing the regional characteristics of central region > eastern region > western region. The latter was mainly due to the fact that the carbon emission intensity of the central region was higher than that of the eastern region in 2005, and the economic development of the central region accelerated from 2005 to 2019, causing the carbon emission intensity to approach the eastern region. The economic development of the western region was relatively slow, and although the carbon emission intensity also declined significantly, the decline was still smaller than that of the eastern and central regions. The western region had a high resistance to industrial structure upgrading, and its scientific and technological level was relatively backward. The higher energy consumption per unit GDP, and economic development lagged behind the eastern and central regions, thus, leading to relatively higher carbon emission intensity. So, compared with 2005, the carbon emission intensity of the eastern and central regions were closer in 2019, and the gap between the western region and the other two regions was more obvious.

#### 4.1.2. Analysis of Regional Differences in Carbon Emissions

(1)Overall differences

In order to explore in-depth the overall regional differences and sources of carbon emissions from urban production energy consumption, this paper used the Dagum Gini coefficient and its decomposition to calculate the overall Gini coefficient of carbon emissions and further decomposed this to measure the Gini coefficients of eastern region, central region and western region.

The overall difference of carbon emission quantities is shown in Table 3. The overall difference of carbon emission quantity fluctuated and decreased, and the overall trend was one of first decreasing and then increasing. Taking 2005 as the base period, the overall difference of carbon emission quantity decreased by 3.69%, and the difference of regional carbon emission quantity decreased. In 2005, the contribution rate of the regional difference was 48.07, indicating a strong difference in carbon emissions between regions. However, in 2019, the difference between regions decreased significantly, and the contribution rate was only 33.99%. At the same time, the contribution rate of the overall regional difference changed into supervariable density > inter-regional difference > intra-regional difference, and the contribution rate of intra-regional difference increased by 2.47%, which was relatively stable. The overall difference of carbon emission intensity is shown in Table 4. The overall difference of carbon emission intensity showed an overall increasing trend during the study period, with the overall difference increasing by 34.31%, and the contribution of inter-regional difference to the overall difference tended to decrease from 2005 to 2019, but still remained the most significant contribution. The contribution rate to the overall difference of carbon emission intensity from 2005 to 2019 was inter-regional difference > intra-regional difference > the supervariable density, evolving into inter-regional difference > the supervariable density > intra-regional difference. From the comparison of the overall Gini coefficient of carbon emission quantity and carbon emission intensity, the overall difference of carbon emission quantity was greater than carbon emission intensity in 2005 and less than carbon emission intensity in 2019, which showed the rapid rise in the differences in the area of carbon emissions intensity, reflecting the enhancement of the imbalance of regional economic development. The specific reason might have been the different levels of technological development and industrial structure.

(2)Intra-regional differences

The intra-regional differences in carbon emission quantity and their evolution trends are shown in Figure 3a. The intra-regional differences in carbon emission quantity in the eastern region were the first among the three major regions, and the intra-regional differences showed an increasing trend during the study period, showing the imbalance of urban production energy consumption in the eastern provinces. The intra-regional difference in carbon emission quantity of the central region and western region showed a differentiated trend.

Since 2012, the intra-regional difference in carbon emission quantity in the central region decreased, while the intra-regional difference in carbon emission quantity in the western region increased, so that the intra-regional difference in the western region was much higher than in the central region by 2019. As shown in Figure 3b, the carbon emission intensity in the eastern region, central region and western region showed a fluctuating upward trend, and the intra-regional difference in carbon emission intensity in the eastern region, central region and western region increased by 34.63%, 35.28% and 45.40%, respectively. During the study period, the differences in the area of carbon emission in the eastern region and western region were similar. In 2019, the western region was slightly higher than the eastern region, while the differences in the central region were always smaller than the eastern region and western region. In general, the difference of carbon emission quantity in the eastern region was much higher than that of carbon emission intensity, the difference of carbon emission quantity in the central region was smaller than that of carbon emission intensity, and the difference of carbon emission quantity in the western region was small.

(3)Inter-regional differences

The inter-regional differences in carbon emissions and their evolution trends are shown in Figure 4a. During the research period, the inter-regional differences of carbon emission quantity in the eastern–western and central–western were very similar. The inter-regional difference of the eastern–western gradually decreased after reaching its maximum value in 2007. The inter-regional difference of the eastern–central continued to increase, and the inter-regional differences of the eastern–central and eastern–western were similar in 2019. The inter-regional difference of the central–western was close to the eastern–central in 2005 and continued to decrease later, and was much smaller than the eastern–central. As shown in Figure 4b, the inter-regional difference in carbon emission intensity had an obvious upward trend, in which the inter-regional difference of the eastern–western was the largest. In 2019 the inter-regional difference of the eastern–western carbon emission intensity reached 0.4514, up 22.08% from 2005, followed by the difference of the central–western, and the difference of the central–western carbon emission intensity rose the most, at 59.96%. The inter-regional difference reached its maximum in 2019. The variation trend of the inter-regional difference of the eastern–central was relatively stable, with a small increase in the study period, and the inter-regional Gini coefficient of eastern–central only increased sharply in 2016, and then decreased and maintained a slow growth.

#### 4.1.3. The Evolution of Carbon Emission Dynamics of Urban Production Energy Consumption

Based on the Dagum Gini coefficient and its decomposition, the overall differences and evolution trends of carbon emission levels were analyzed, and the relative differences of regions were identified. In order to further study the dynamic distribution characteristics of carbon emission levels and absolute regional differences, this paper used Kernel density estimation to study the overall pattern and dynamic evolution of carbon emission distribution in China and each region. Matlab software was used to draw a three-dimensional perspective view of carbon emission levels in China and each of its regions from 2005 to 2019.

(1)The evolution of carbon emission quantity dynamics

As shown in Figure 5a, the center of the peak of the kernel density curve of carbon emission quantity shifted left and then right from 2005 to 2019, and, compared with 2005, the center of the kernel density curve shifted right and the peak decreased significantly in 2019. However, the decline gradually slowed down and a new wave appeared at the right end. The main peak gradually changed from a sharp peak to a broad peak. As shown in Figure 5b–d, there was a slight rightward shift in the center of the peak of carbon emission quantity in the eastern region from 2005 to 2019, while there was a significant rightward shift in the central region and a significant leftward shift in the western region. The peaks in the eastern region, central region and western region all showed a decreasing trend during the study period and gradually evolved from sharp peaks to broad peaks. The eastern region and western region were always single-peaked, while the western region had a side peak to the right of the main peak from 2008 to 2019, but the peak was lower and gradually became flat. The main peak of the kernel density curve of carbon emission quantity in the three regions during the study period was western region > central region > eastern region.

The above analysis showed that the imbalance of carbon emission quantities in China deepened, and there was a trend of polarization, with large differences between regions and an increasing trend of differences between provinces within each region. The largest differences between provinces were within the eastern region, followed by the central region, and with polarization within the western region, but the polarization was gradually decreasing.

(2)The evolution of carbon emission intensity dynamics

As shown in Figure 6a, the overall carbon emission intensity kernel density curve from 2005–2019 showed a right trailing phenomenon. The peak showed an upward trend, the center of the peak first shifted right and then shifted significantly left during the study period, and evolved from single to multiple peaks. There were two side peaks at the right side of the main peak in 2019, but the side peaks were smaller and the trend was flatter. As shown in Figure 6b–d, the kernel density curves of carbon emission intensity in the eastern region, central region and western region all evolved from single to multiple peaks with a right trailing phenomenon from 2005 to 2019. Among them, the eastern region had a side peak on the right side of the main peak in 2019, and the central and western regions had two side peaks on the right side of the main peak in 2019. The main and side peaks in the eastern region increased significantly during the study period, and the center of the main peak did not have a significant shift. The main and side peaks in the central region increased significantly, but the center of the main peak had a significant left shift. The main peak in the western region increased significantly, while the side peaks had strong volatility and the center of the main peak had a significant left shift. Overall, the main peak of the kernel density curve of carbon emission intensity was central region > eastern region > western region.

The above analysis showed that the imbalance of carbon emission intensity of urban production energy consumption in the country weakened, but the deepening of the polarization phenomenon indicated an obvious gradient effect. Compared with the eastern region and western region, the central region had the weakest imbalance, followed by the eastern region, and the western region had the strongest imbalance. The emergence of side peaks in the eastern region and central region also showed a certain polarization phenomenon. There was an obvious gradient effect, namely most provinces’ carbon intensity decreased obviously and became gradually numerically close, but there were still some provinces where the decrease in carbon emission intensity was small and there was a large gap with the carbon emission intensity in most provinces, so the phenomenon of multi -polarization was generated.

### 4.2. Carbon Emission Classification of Urban Production Energy Consumption

The carbon emission classification of urban production energy consumption inter-provincially is shown in Table 5. From 2005 to 2019, carbon emissions evolved from high-low category to high-high category in Liaoning, from low-low category to high-high category in Xinjiang, and from low-low category to low-high category in Heilongjiang, indicating that carbon emission intensity in Liaoning and Heilongjiang, and carbon emission quantity and intensity in Xinjiang jumped from below-average to above-average. Hubei evolved from high-high category to high-low category, Jilin, Guizhou and Yunnan evolved from low-high category to low-low category, and Hunan evolved from high-low category to low-low category, indicating that the carbon emission intensity of Hubei, Jilin, Guizhou and Yunnan and the carbon emission quantity of Hunan all jumped from above-average to below-average. The carbon emission classification shows that the number of provinces in China at low carbon emission quantity and low carbon emission intensity increased in 2019 compared to 2005, with the number of provinces reaching 50%. Most provinces produced large changes in carbon emission categories during the study period, also showing differences in their production energy consumption levels. Provinces with relatively high carbon emission quantity or relatively high carbon emission intensity should start from different paths to reduce emissions. For provinces in the high-low category, they should mainly start from the perspective of energy use, focusing on the development and utilization of clean energy and reducing the use of fossil energy, such as coal. For provinces in the low-high category, they should mainly start from the technical level, strengthen the level of scientific and technological innovation, and improve the efficiency of energy use. For provinces in the high-high category, they should consider both energy consumption and technological innovation.

### 4.3. Analysis of Driving Factors of Carbon Emission from Production Energy Consumption

In the previous section, the analysis of carbon emission levels of urban production energy consumption inter-provincially found that there were significant spatial distribution characteristics. In order to further explore the strategies to reduce the carbon emission levels of production energy consumption, it was necessary to conduct an in-depth study of the driving mechanism. By calculating the global Moran’s I index of carbon emissions for spatial correlation analysis (Table 6), it was found that the quantity and intensity of carbon emissions from 2005 to 2019 had significant spatial correlation. Therefore, this paper analyzed the driving factors of carbon emission quantity and carbon emission intensity using a spatial econometric model. To reduce the possible heteroskedasticity and fluctuation effects of the variables, both explanatory and explained variables were logarithmically treated. LM tests [70] of each explanatory variable, using the spatial econometric model, found that both the spatial error term and the spatial lag term of carbon emission quantity and carbon emission intensity were significant (Table 7). Further, Wald and LR tests were conducted for the spatial Durbin model of carbon emission quantity and carbon emission intensity, and it was found that the spatial Durbin model of carbon emission quantity and carbon emission intensity could not degenerate into a spatial error and spatial lag model. Finally, the Hausman test was performed on the model and the fixed-effect model was selected, and the model was finally determined by comparing the intra-group R-square and AIC and BIC criteria. The model selection is shown in Table 8.

#### 4.3.1. Analysis of the Driving Factors of Carbon Emissions from Urban Production Energy Consumption from a National Perspective

As shown in Table 9, the carbon emission quantity from production energy consumption was influenced by multiple factors, among which foreign direct investment and scientific and technological innovation had a significant inhibiting effect on the carbon emission quantity. Foreign direct investment had a significant suppressing effect on carbon emission quantity in both local and neighboring regions, while scientific and technological innovation had a significant suppressing effect mainly on local carbon emission quantity. Foreign direct investment and scientific and technological innovation may control the amount of carbon emissions by promoting technological progress and, thus, improving the efficiency of energy use, as well as promoting the upgrading of industrial structure. The change in industrial structure (the increase of the value added of secondary industry as a share of GDP), the increase of per capita disposable income, urbanization rate and cultural level had significant promoting effects on carbon emission quantity. The change of industrial structure had significant promoting effects on carbon emission quantity in both local and neighboring regions. The effect of per capita disposable income on local carbon emission quantity was significantly positive, and the urbanization rate only had a significant positive total effect. The effect of cultural level only had a significant positive effect on carbon emission quantity in neighboring regions. The upgrading of industrial structure (increasing the ratio of third, or tertiary, industry to secondary industry) can make industries develop to a higher level and reduce their dependence on resources, thus, reducing carbon emission quantity. Secondly, the upgrading of industrial structure can also significantly reduce carbon emission quantity by promoting technological innovation [71], so the change of industrial structure in this study could lead to an increase in carbon emission quantity. On the one hand, the level of urbanization might reduce carbon emission quantity by improving the energy structure and technology level, but, on the other hand, the urbanization rate increases the level of residential consumption, and the effect is higher than the improvement of energy structure and technology level. The increase of residential consumption level further leads to increase in carbon emissions quantity, so the urbanization rate had a significant positive effect on the increase of carbon emission quantity [72]. The increase of per capita disposable income reflected the increase of the consumption capacity of the population, which, in turn, promoted the production efforts of the production industries and, therefore, led to more carbon emission quantity from energy consumption.

Carbon emission intensity was also affected by multiple factors, as shown in Table 10. The direct, indirect and total effects of foreign direct investment and scientific and technological innovation on carbon emission intensity were significantly negative, indicating that foreign direct investment and scientific and technological innovation could effectively suppress the carbon emission intensity of local and neighboring areas. The change of industrial structure had a significant inhibitory effect on local carbon emission intensity, but significantly increased the carbon emission intensity of neighboring areas, because the secondary industry consumed more local energy, but also promoted the economic level, so the effect on local areas was economic enhancement > carbon emission, but the effect on neighboring areas was carbon emission > economic enhancement. So, the effect on carbon emission intensity was heterogeneity. Per capita disposable income reflects the improvement of local economic level, so the increase of per capita disposable income could significantly suppress local carbon emission intensity, but the indirect effect and total effect were not significant. Energy consumption structure and greening coverage of built-up areas had a significant direct contribution effect on carbon emission intensity, and cultural level had a significant direct suppression effect. The dependency ratio of elderly population and cultural level had significant spatial spillover effects that significantly increased carbon emission intensity in neighboring provinces, where the total effect of dependency ratio of elderly population on carbon emission was significantly positive. This showed that the increase of dependency ratio of elderly enhanced the pressure on the young population, leading to brain drain, enterprise migration, etc., and slow economic growth, which, in turn, led to the increase of carbon emission intensity [62]. When the spatial factor was not considered, the increase of urbanization rate could reduce the carbon emission intensity of energy consumption [73], but this study found that the inhibitory effect of urbanization rate on the carbon emission intensity of energy consumption of provincial production was not significant after considering the spatial spillover effect, and the urbanization rate still mainly had a significant effect on carbon emission quantity. The growth of tertiary industry value added as a share of GDP and economic growth could suppress carbon emission intensity, and there was also a significant causal relationship between urbanization rate and economic growth. Therefore, a high-quality level of urbanization is essential for energy saving and emission reduction and reducing carbon emission intensity [74].

#### 4.3.2. Analysis of the Driving Factors of Carbon Emissions from Urban Production Energy Consumption in a Regional Perspective

The results of the analysis of carbon emission driving factors of urban production energy consumption in the three major regions of China are shown in Table 11 and Table 12. The change in industrial structure could significantly enhance carbon emission quantity in the eastern and central regions and reduce carbon emission quantity in the western region, among which the direct effect of the change in industrial structure on the central region was not significant, but had a significant positive effect on the indirect and total effects. The direct, indirect and total effects of industrial structure on the western region all significantly showed that the increase in the proportion of value added in secondary industry could reduce carbon emission quantity in the western region. There was no significant spatial spillover effect of foreign direct investment on the eastern and western regions, but it could significantly raise the local carbon emission quantity in the western region and reduce the carbon emission quantity in the neighboring provinces, which, in turn, had a significant inhibitory effect on the overall carbon emission quantity at the provincial level. Per capita disposable income had a more significant effect on carbon emission quantity in the eastern region, which could increase local carbon emission quantity and suppress carbon emission quantity in neighboring provinces, but its contribution to carbon emission quantity was stronger, thus making overall carbon emission quantity higher, which was consistent with the results of the China-wide regression. The spatial spillover effect of the change in energy consumption structure (rising share of coal consumption in energy consumption) was not significant, but could significantly raise carbon emission quantity in the eastern, central and western regions. The increase in urbanization rate and cultural level could reduce carbon emission quantity in the western region, but it mainly had a significant inhibitory effect on local carbon emission quantity. The opposite effect of the increase in urbanization rate for the eastern and central regions was mainly due to the difference in the development stages of urbanization rate between the eastern, central and the west regions during the study period, and, therefore, might have had different effects on the carbon emission quantity.

As can be seen from Table 11 and Table 13, the change in industrial structure had significant direct and indirect effects on the eastern region, but the total effect was not significant, that is, it could significantly enhance the local carbon emission intensity and suppress the carbon emission intensity of neighboring regions, but the direct and indirect effects interacted with each other, so that the effect of the change in industrial structure on the overall inter-provincial carbon emission intensity was not significant. At the same time, the change in industrial structure could significantly suppress the carbon emission intensity in the western region, and had a strong spatial spillover effect. The increase of foreign direct investment had significant spatial spillover and total effects on carbon emission intensity in the eastern and western regions, but the difference was that foreign direct investment raised the carbon emission intensity in the eastern region and suppressed the carbon emission intensity in the western region. The total effect of rising per capita disposable income was negative and significant for the eastern, central and western regions, which could suppress carbon emission intensity, with per capita disposable income having comparable suppression effects on carbon emission intensity of local and neighboring provinces in the eastern region, and tending to suppress carbon emission intensity of neighboring provinces more for the central region. The change in energy consumption structure had a significant positive effect on the local provincial carbon emission intensity in the eastern, central and western regions, and the total effect on the carbon emission intensity in the central and western regions was significantly positive, and had a significant spatial spillover effect on the western region and a negative effect on the neighboring provinces in the eastern region. Therefore, the direct and indirect effects in the eastern provinces had a mutual offset, thus, leading to an insignificant total effect. The dependency ratio of elderly population had a significant effect on the western region, which would significantly increase the carbon emission intensity of local and neighboring provinces. The direct and total effects of increasing green coverage of built-up areas on carbon emission intensity in the central region were significantly positive, with no significant spatial spillover effect. The increase in urbanization rate had no significant effect on the central region, but the effect on the eastern region was opposite to the western region. The increase in urbanization rate increased the carbon emission intensity of the neighboring provinces in the eastern region, but was able to suppress the carbon emission intensity of the local provinces in the western region. The spatial spillover effect and the total effect of cultural level in the central region were significantly positive, while the increase of cultural level in the western region could effectively reduce the carbon emission intensity of the province and neighboring provinces. The improvement of scientific and technological innovation had a negative spatial spillover effect and total effect on carbon emission intensity in the western region, i.e., it could significantly reduce carbon emission intensity in the western region.

#### 4.3.3. Test for Robustness

In order to test the feasibility and robustness of the research results, this article adopted a replacement indicator method for a robustness test. The per capita disposable income of important indicators was replaced by per capita consumption expenditure. The robustness test results are shown in the Table 14. The coefficient of the per capita consumption expenditure effect could promote carbon emission quantity at a significant level of 1%, reducing carbon emission intensity at a significant level of 1%, which was the same as the result of the per capita disposable income. The coefficients of indirect effects and total effects were similar to the coefficients of per capita disposable income. In addition, other important indicators were the same as the return of regression after replacing the per capita consumption expenditure. Therefore, urban living standards could promote the carbon emission quantity from production energy consumption and inhibit the carbon emission intensity from production energy consumption. It also showed that the carbon emission quantity and carbon emission intensity from production energy consumption were inseparable from the impact of urban energy consumption level, urban population size and urban development level.

## 5. Conclusions

This study measured the carbon emission levels (carbon emission quantity and carbon emission intensity) of urban production energy consumption in inter-provincial areas in China based on data from 2005–2019. The study measured and decomposed the overall regional differences in carbon emission levels using the Dagum Gini coefficient, analyzed the dynamic characteristics of the distribution of carbon emission levels and absolute regional differences using the Kernel density estimation method, and used a spatial econometric model to analyze the driving factors of carbon emission levels in the whole region and three regions (eastern region, central region and western region). The main research findings are as follows.

First, the carbon emission quantity increased year by year during the study period, but the growth rate decreased, and the carbon emission intensity had an obvious decreasing trend. Besides this, the carbon emission quantity and carbon emission intensity showed obvious spatial differences, i.e., the carbon emission quantity showed the characteristic of eastern region > central region > western region, while the carbon emission intensity, on the contrary, showed the characteristic of western region > central region > eastern region, and the spatial characteristics were more obvious as time went on.

Second, the overall difference of carbon emission quantity showed a decreasing trend. The intra-regional difference of carbon emission quantity evolved from eastern region > central region > western region to eastern region > western region > central region, and the overall difference of carbon emission intensity showed an increasing trend, and the intra-regional difference of carbon emission intensity was always western region > eastern region > central region. The imbalance of carbon emission quantity increased in the country, and the imbalance of carbon emission quantity increased in different degrees in the eastern region, central region and western region. There was a significant gradient effect in the carbon emission intensity, and there was also a trend of multi -level differentiation in the provinces in various regions. From the perspective of carbon emission classification, the inter-provincial carbon emission classification had some changes from 2005 to 2019, but most of the carbon emission provinces were in the low-low category.

Third, carbon emission level was influenced by multiple factors, among which carbon emission quantity was mainly influenced by industrial structure, per capita disposable income, energy consumption structure, urbanization rate, cultural level, scientific and technological innovation. Carbon emission intensity was mainly influenced by industrial structure, foreign direct investment, energy consumption structure, the dependency ratio of elderly population and scientific and technological innovation. Different regions had heterogeneity, and the level of driving factors was different. Industrial structure, per capita disposable income, energy consumption structure, urbanization rate were the important driving factors of regional carbon emissions.

## 6. Policy Suggestions

Based on the above findings, this paper proposes the following recommendations to reduce carbon emissions from urban production energy consumption in China, to promote the achievement of carbon peaking and carbon neutral goals, and to realize green and sustainable economic and social development.

First, on a national scale, the energy consumption structure has more room for transformation. On the one hand, China’s natural resource profile of “coal-rich and oil-poor” has influenced the energy consumption structure, so China’s current production energy consumption is still coal-based fossil energy, which leads to a large amount of CO_2_ emissions [75]. On the other hand, the energy consumption structure also depends on the industrial structure, and as a large industrial country, China’s industrial sector consumes relatively more energy. Therefore, we should optimize the energy consumption structure and industrial structure. Industrial structure is the core driving force for the development of low-carbon cities [76]. Adjusting the industrial structure, accelerating the transformation and upgrading of high-energy-consuming industries, and then adjusting the energy consumption structure could effectively reduce urban carbon emissions. In addition, strengthening scientific and technological innovations, especially the development and utilization of clean energy, improving the construction of new energy systems with clean power as the main body, strengthening the development of wind power and nuclear power, reducing the energy consumption of industrial products, and reducing the dependence on fossil fuels, such as coal, and realizing a low-carbon economy would all be beneficial. From a regional perspective, the energy consumption structure has a heterogeneous impact on the region, and more attention should be paid to the transformation of the energy consumption structure in the central region and western region. The change of industrial structure also has an important impact on economic development, and “low carbon” can be used to force traditional industries to upgrade and promote the development of new industries, but it should also be considered that industrial restructuring should go hand in hand with economic development, and excessive pursuit of economic growth or excessive emphasis on low carbon is not conducive to the long-term development of the country. Balance of the low carbon economy must be sought.

Secondly, the regional differences should be balanced to achieve dual balance between carbon emission quantity and carbon emission intensity. The regional differences in carbon emission quantity and carbon emission intensity of urban production energy consumption reflect imbalance in intra- and inter-regional economic development and energy utilization. The eastern region as a whole has a relatively developed economy and higher technology level, but its internal imbalance is also stronger. Therefore, important provinces in the eastern region, such as Shanghai and Zhejiang, should actively play the role of central cities to strengthen the economic and technological radiation to the surrounding areas, improving the economic levels of the surrounding provinces while reducing the differences in the level of carbon emissions of production energy within the region. The eastern region should give full play to its advantages in high and new technology, and transport technical talents to the central region and western region, so as to provide important support for the industrial development and upgrading of the central region and western region. The difference of carbon emission level and economic development within the central region is smaller, therefore, as an important hub between the eastern region and western region, the central provinces should play a good role as a mediator, while maintaining a stable and positive development. The western region should pay more attention to economic construction and low-carbon development of secondary and tertiary industries, while industrial structure transformation is crucial to strengthen the upgrading of low-end manufacturing industries to high-end manufacturing industries, accelerate the transfer from labor and capital-intensive industries to high-tech-intensive ones, and narrow the economic differences with the eastern region and central region, so as to reduce the carbon emission intensity of production.

Finally, the construction of new urbanization should be strengthened. The increase of urbanization rate has significant heterogeneity for regions, and the urbanization rate is higher in the eastern region and central region at the present stage. Further increasing the level of urbanization rate will raise the level of production of carbon emissions, so the transformation of new urbanization should be accelerated, and new urbanization should play an important role in revitalizing the economy, while being a key step in the coordinated development of economic and ecological environments [77]. Therefore, new-type urbanization should be strengthened, especially in the eastern region and central region, to eliminate the high pollution and carbon emissions brought by traditional urbanization construction. Specifically, the new urbanization should, on the one hand, strengthen the urbanization of population, not only the migration of population from rural to urban areas, but more importantly, the overall improvement in production and living consumption levels, and, on the other hand, strengthen the rationalization of planning so that development is not at the expense of environment Development of special industries according to local conditions, etc. should be encouraged.

## Figures and Tables

**Figure 1 ijerph-19-12441-f001:**
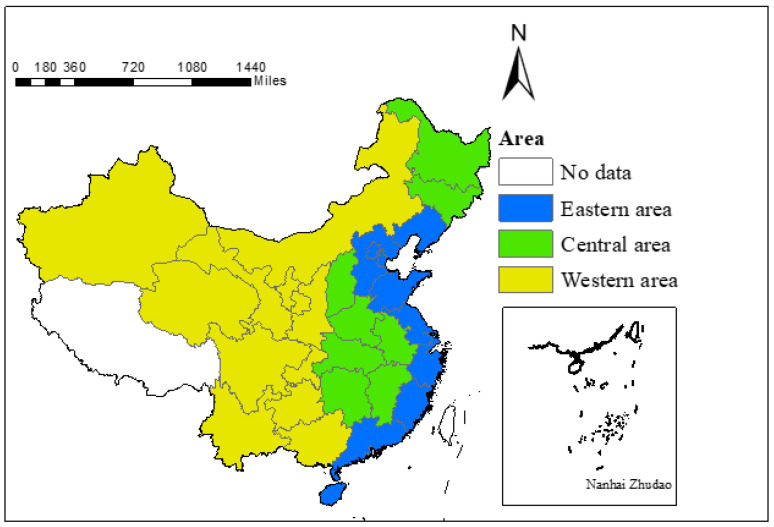
Study area.

**Figure 2 ijerph-19-12441-f002:**
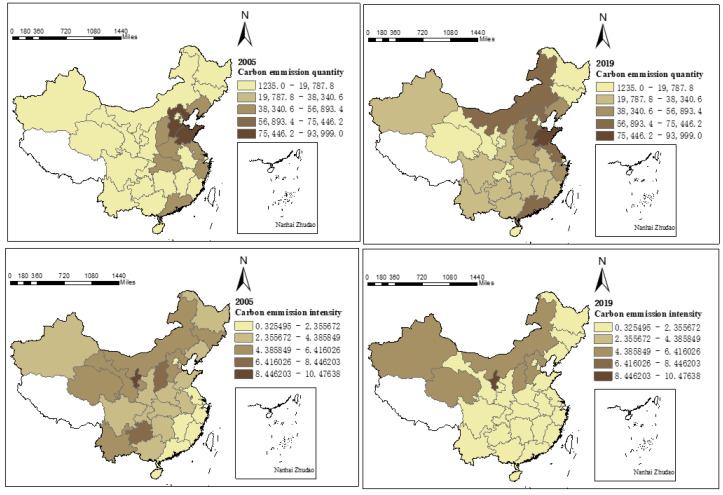
Spatial distribution of carbon emissions from production energy consumption in 30 provinces of China.

**Figure 3 ijerph-19-12441-f003:**
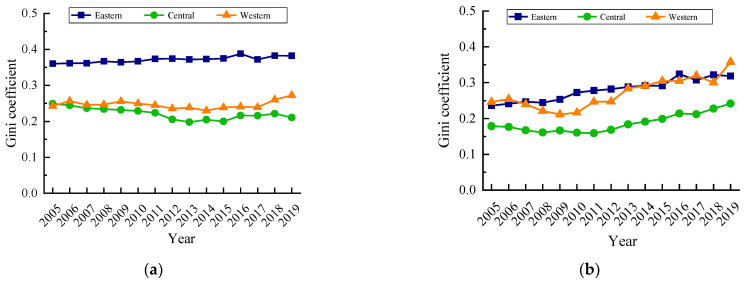
Intra-regional Gini coefficient of carbon emissions of urban production energy consumption from 2005 to 2019. (**a**) Carbon emission quantity. (**b**) Carbon emission intensity.

**Figure 4 ijerph-19-12441-f004:**
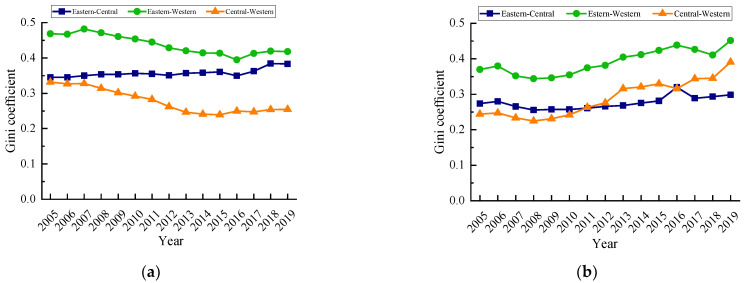
Inter-regional Gini coefficient of carbon emissions of urban production energy consumption from 2005 to 2019. (**a**) Carbon emission quantity. (**b**) Carbon emission intensity.

**Figure 5 ijerph-19-12441-f005:**
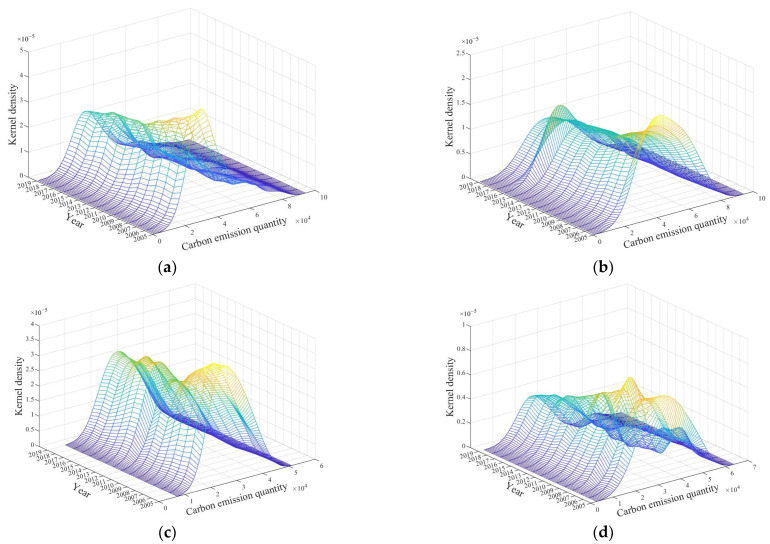
Kernel density estimation of national and regional carbon emission quantity from 2005 to 2019. (**a**) National. (**b**) Eastern region. (**c**) Central region. (**d**) Western region.

**Figure 6 ijerph-19-12441-f006:**
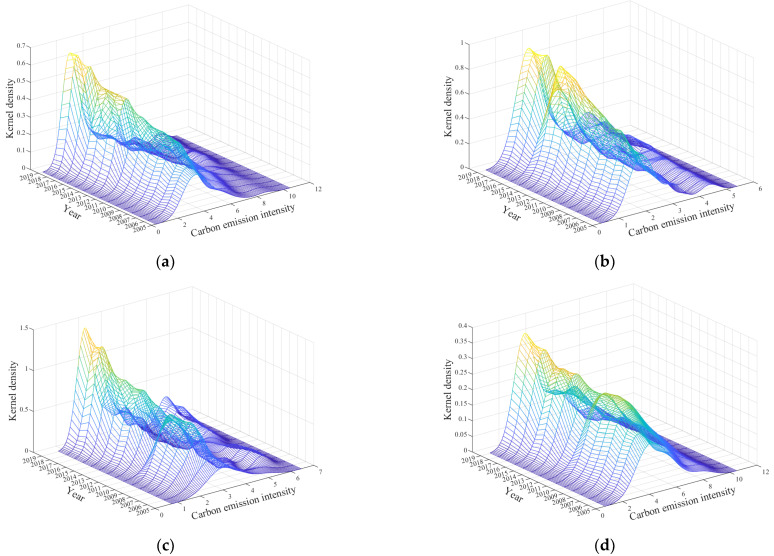
Kernel density estimation of national and regional carbon emission intensity from 2005 to 2019. (**a**) National. (**b**) Eastern region. (**c**) Central region. (**d**) Western region.

**Table 1 ijerph-19-12441-t001:** Abbreviations, definitions, and data sources for variables.

Variable Name		Abbreviation	Definition	Source
Carbon emissions from urban production energy consumption	Carbon emission quantity	CEQ (million tons)	Carbon emission quantity from energy consumption in the production process of industry, raw materials and materials, construction, transportation, warehousing, postal services, wholesale, retail, accommodation and catering.	China Energy Statistical Yearbook
	Carbon emission intensity	CEI (million tons/billion yuan)	Carbon emission quantity per unit of value added in secondary and third industries.	China Energy Statistical Yearbook, China Statistical Yearbook [69]
Urban economic level	Industrial structure	IS (%)	The proportion of agriculture, industry and services in a country’s economic structure.	China Statistical Yearbook
	Foreign direct investment	FDI (billion yuan)	The act of direct investment in China by foreign enterprises, economic organizations or individuals using cash, material goods and technology in accordance with relevant Chinese policies and regulations.	China Statistical Yearbook
Living standard of urban residents	Per capita disposable income	PCDI (yuan)	The sum of final consumption expenditure and savings available to residents, that is, the income available to residents for discretionary use.	China Statistical Yearbook
	Per capita consumption expenditure	PCCE (yuan)	The total expenditure of residents to meet the daily consumption of the family, including the purchase of goods and service consumption expenditure.	China Statistical Yearbook
Urban energy consumption level	Energy consumption structure	ECS (%)	The quantity of each type of energy consumed by each sector of the national economy in a certain period and its proportion in the total energy consumption, or the energy consumption and its proportion according to the consumption sector.	China Energy Statistical Yearbook
Urban Population Size	Dependency ratio of elderly population	DREP (%)	The ratio of the middle and old part of the population to the number of working-age people.	China Statistical Yearbook
	Population density	PD (persons/km^2^)	The number of people per unit of land area.	China Statistical Yearbook
Urban development level	Green coverage rate of built-up area	GCR (%)	The percentage of the green coverage area in the urban built-up area.	China Statistical Yearbook
	Urbanization rate	UR (%)	Central urban area, county (city, district) and administrative town, where included in the urban construction planning and urban construction, have been extended to the township, neighborhood committee and village committee and have realized water, electricity, road; “three links”.	China Statistical Yearbook
	Cultural level	CL (years)	An important indicator of the population quality of a country. It marks the popularization and development degree of a country’s culture and education.	China Statistical Yearbook
	Scientific and technological innovation	STI (million yuan)	Industrial enterprises are used for specific activities in scientific and technological innovation and development.	China Statistical Yearbook

**Table 2 ijerph-19-12441-t002:** Descriptive statistics of variables.

Variable Name	Mean	Std. D.	Min	Max
CEQ (million tons)	26,048.74	17,899.20	1235.00	93,999.00
CEI (million tons/billion yuan)	2.36	1.54	0.33	10.48
IS (%)	0.43	0.08	0.16	0.62
FDI (billion yuan)	464.32	503.71	0.31	2467.27
PCDI (yuan)	23,790.70	11,698.71	8013.00	73,849.00
PCCE (yuan)	16,788.26	7642.61	5960.00	48,272.00
ECS (%)	0.43	0.16	0.01	0.76
DREP (%)	13.47	2.98	7.40	23.80
PD (persons/km^2^)	2734.90	1266.09	189.00	6307.00
GCR (%)	37.72	4.59	23.50	49.10
UR (%)	54.08	13.83	26.87	89.60
CL (years)	8.81	1.01	6.38	12.78
STI (million yuan)	2,489,551.00	5,946,871.00	5349.37	57,000,000.00

**Table 3 ijerph-19-12441-t003:** Gini coefficient of carbon emission quantity from urban production energy consumption.

Year	Overall	Intra-Regional		Inter-Regional		Supervariable Density	
		Source	Contribution Rate (%)	Source	Contribution Rate (%)	Source	Contribution Rate (%)
2005	0.3628	0.1041	28.70	0.1744	48.07	0.0843	23.23
2006	0.3625	0.1050	28.97	0.1707	47.07	0.0868	23.95
2007	0.3668	0.1041	28.38	0.1818	49.57	0.0809	22.05
2008	0.3636	0.1049	28.85	0.1742	47.90	0.0845	23.25
2009	0.3588	0.1050	29.27	0.1666	46.42	0.0872	24.31
2010	0.3556	0.1047	29.43	0.1605	45.12	0.0905	25.44
2011	0.3513	0.1047	29.80	0.1506	42.88	0.0960	27.32
2012	0.3403	0.1024	30.10	0.1406	41.32	0.0973	28.58
2013	0.3365	0.1022	30.37	0.1348	40.07	0.0995	29.56
2014	0.3336	0.1017	30.49	0.1297	38.86	0.1023	30.65
2015	0.3345	0.1026	30.68	0.1267	37.87	0.1052	31.45
2016	0.3304	0.1048	31.71	0.1082	32.74	0.1175	35.55
2017	0.3364	0.1030	30.62	0.1230	36.57	0.1104	32.81
2018	0.3495	0.1084	31.03	0.1227	35.09	0.1184	33.88
2019	0.3494	0.1089	31.17	0.1187	33.99	0.1217	34.85

**Table 4 ijerph-19-12441-t004:** Gini coefficient of carbon emission intensity from urban production energy consumption.

Year	Overall	Intra-Region		Inter-Regional		Supervariable Density	
		Source	Contribution Rate (%)	Source	Contribution Rate (%)	Source	Contribution Rate (%)
2005	0.2777	0.0779	28.07	0.1539	55.43	0.0458	16.51
2006	0.2838	0.0797	28.07	0.1565	55.13	0.0477	16.79
2007	0.2681	0.0768	28.66	0.1396	52.06	0.0517	19.28
2008	0.2587	0.0731	28.27	0.1388	53.67	0.0467	18.06
2009	0.2608	0.0730	27.97	0.1411	54.10	0.0468	17.93
2010	0.2683	0.0757	28.20	0.1433	53.41	0.0493	18.39
2011	0.2853	0.0817	28.63	0.1515	53.11	0.0521	18.27
2012	0.2920	0.0829	28.39	0.1562	53.49	0.0529	18.12
2013	0.3170	0.0919	28.98	0.1665	52.51	0.0587	18.51
2014	0.3232	0.0940	29.08	0.1703	52.69	0.0589	18.23
2015	0.3325	0.0969	29.16	0.1765	53.07	0.0591	17.78
2016	0.3456	0.1030	29.81	0.1805	52.23	0.0621	17.96
2017	0.3420	0.1029	30.10	0.1695	49.56	0.0695	20.34
2018	0.3380	0.1010	29.89	0.1555	46.00	0.0815	24.11
2019	0.3729	0.1131	30.34	0.1812	48.58	0.0786	21.08

**Table 5 ijerph-19-12441-t005:** Carbon emission classification from urban production energy consumption.

Classification		High-High	High-Low	Low-High	Low-Low
Carbon emissions of urban production energy consumption	2005	Hebei, Shanxi, Inner Mongolia, Hubei	Liaoning, Jiangsu, Zhejiang, Shandong, Henan, Hunan, Guangdong	Jilin, Guizhou, Yunnan, Gansu, Qinghai, Ningxia	Beijing, Tianjin, Heilongjiang, Shanghai, Anhui, Fujian, Jiangxi, Guangxi, Hainan, Chongqing, Sichuan, Shaanxi, Xinjiang
	2019	Hebei, Shanxi, Inner Mongolia, Hubei, Liaoning, Xinjiang	Jiangsu, Zhejiang, Shandong, Henan, Hubei, Guangdong	Heilongjiang, Gansu, Qinghai, Ningxia	Beijing, Tianjin, Jilin, Shanghai, Anhui, Fujian, Jiangxi, Hunan, Guangxi, Hainan, Chongqing, Sichuan, Guizhou, Yunnan, Shaanxi

**Table 6 ijerph-19-12441-t006:** Moran’s I index of contribution rate of CEQ and CEI from 2005 to 2019.

Year	I		E(I)		SD(I)		Z-Value		*p*-Value	
	CEQ	CEI	CEQ	CEI	CEQ	CEI	CEQ	CEI	CEQ	CEI
2005	0.266	0.230	−0.034	−0.034	0.119	0.115	2.515	2.294	0.006	0.011
2006	0.264	0.211	−0.034	−0.034	0.120	0.117	2.490	2.101	0.006	0.018
2007	0.263	0.231	−0.034	−0.034	0.120	0.118	2.491	2.250	0.006	0.012
2008	0.244	0.276	−0.034	−0.034	0.119	0.119	2.339	2.609	0.010	0.005
2009	0.236	0.321	−0.034	−0.034	0.119	0.120	2.262	2.969	0.012	0.001
2010	0.225	0.315	−0.034	−0.034	0.120	0.120	2.171	2.916	0.015	0.002
2011	0.217	0.314	−0.034	−0.034	0.120	0.118	2.098	2.943	0.018	0.002
2012	0.194	0.349	−0.034	−0.034	0.119	0.119	1.916	3.229	0.028	0.001
2013	0.191	0.358	−0.034	−0.034	0.119	0.117	1.891	3.354	0.029	0.000
2014	0.189	0.361	−0.034	−0.034	0.119	0.116	1.873	3.392	0.031	0.000
2015	0.181	0.394	−0.034	−0.034	0.119	0.115	1.811	3.727	0.035	0.000
2016	0.251	0.404	−0.034	−0.034	0.118	0.116	2.423	3.777	0.008	0.000
2017	0.164	0.400	−0.034	−0.034	0.120	0.114	1.653	3.805	0.049	0.000
2018	0.155	0.381	−0.034	−0.034	0.119	0.112	1.591	3.705	0.056	0.000
2019	0.144	0.396	−0.034	−0.034	0.119	0.113	1.499	3.825	0.067	0.000

**Table 7 ijerph-19-12441-t007:** LM test results.

LM Test		
	Statistics	
	CEQ	CEI
Lagrange multiplier	14.276 (0.000)	63.825 (0.000)
Robust Lagrange multiplier	8.043 (0.005)	10.713 (0.000)
Lagrange multiplier	19.666 (0.000)	72.954 (0.000)
Robust Lagrange multiplier	13.433 (0.000)	19.842 (0.000)

**Table 8 ijerph-19-12441-t008:** Model selection results.

Explained Variables	National	Eastern Region	Central Region	Western Region
Carbon emission quantity	Time, individual double fixed effects spatial Durbin model	Time, individual double fixed effects spatial Durbin model	Time, individual double fixed effects spatial Durbin model	Time, individual double fixed effects spatial Durbin model
Carbon emission intensity	Individual fixed effects spatial Durbin model	Individual fixed effects spatial Durbin model	Time, individual double fixed effects spatial Durbin model	Time, individual double fixed effects spatial Durbin model

**Table 9 ijerph-19-12441-t009:** Regression results of the region-wide sample of carbon emissions from urban production energy consumption.

Variable	lnCEQ	lnCEI
lnIS	0.1406	−0.2389 ***
lnFDI	−0.0435 ***	−0.0378 ***
lnPCDI	1.6895 ***	−0.4153 ***
lnECS	0.1216 ***	0.1575 ***
lnDREP	0.0293	−0.0018
lnPD	0.0134	0.0012
lnGCR	0.2361 **	0.1935 **
lnUR	0.2419	0.1908
lnCL	−0.1314	−0.5789 **
lnSTI	−0.0401 ***	−0.0634 ***
Spatial		
ρ	0.1233 *	0.3134 ***
Variance		
σ^2^ _e	0.0090 ***	0.0107 ***
Time effect	Yes	No
Individual effect	Yes	Yes
R^2^	0.7793	0.8990
AIC	−799.0385	−709.7291
BIC	−708.635	−619.3257
N	450	450

Note: *, **, *** denote significant levels of 10%, 5%, and 1% levels, respectively.

**Table 10 ijerph-19-12441-t010:** Decomposition of carbon emission intensity effect of urban production energy consumption.

Variable	CEQ			CEI		
	Direct Effect	Indirect Effect	Total Effect	Direct Effect	Indirect Effect	Total Effect
lnIS	0.1458 *	0.3569 *	0.5027 *	−0.1944 **	0.7738 ***	0.5794 *
lnFDI	−0.0457 ***	−0.1224 ***	−0.1681 ***	−0.0468 ***	−0.1658 ***	−0.2126 ***
lnPCDI	1.6967 ***	−0.0970	1.5997 ***	−0.4014 ***	0.1865	−0.2150
lnECS	0.1199 ***	−0.0762	0.0437	0.1578 ***	−0.0011	0.1567 *
lnDREP	0.0301	0.0538	0.0839	0.0168	0.3042 **	0.3210 **
lnPD	0.0150	0.0205	0.0356	−0.0004	−0.0403	−0.0407
lnGCR	0.2289 **	−0.2675	−0.0386	0.1944 *	0.0096	0.2040
lnUR	0.2431	0.2733	0.5165 *	0.1528	−0.5059	−0.3531
lnCL	−0.0559	1.9291 ***	1.8732 ***	−0.4873 *	1.2247 ***	0.7374
lnSTI	−0.0402 ***	−0.0182	−0.0583 *	−0.0675 ***	−0.0760 **	−0.1435 ***

Note: *, **, *** denote significant levels of 10%, 5%, and 1% levels, respectively.

**Table 11 ijerph-19-12441-t011:** Regression results of carbon emissions from urban production energy consumption by region sample.

Variable	Eastern Region	Central Region	Western Region	Eastern Region	Central Region	Western Region
	lnCEQ	lnCEQ	lnCEQ	lnCEI	lnCEI	lnCEI
lnIS	0.2926 *	0.2638	−0.3257 **	0.4619 ***	−0.7836 ***	−0.7022 ***
lnFDI	−0.0136	−0.0547 **	0.0221 *	0.0167	−0.0349	0.0015
lnPCDI	1.1254 ***	1.2471 **	0.0318	−0.4945 ***	−0.0839	−0.5072 *
lnECS	0.0658 **	0.2046 ***	0.4426 ***	0.0817 **	0.3085 ***	0.4751 ***
lnDREP	0.1096	−0.2584 **	−0.1035	0.2065 ***	−0.1471	−0.0281
lnPD	0.0222	0.0351	−0.0260	−0.0671	−0.0122	−0.0261
lnGCR	0.1067	0.6304 ***	0.1140	−0.1242	0.9837 ***	0.0339
lnUR	0.3179	0.0128	−2.2996 ***	0.1686	0.1097	−2.3494 ***
lnCL	0.1700	−0.2085	−0.6438 **	−0.2728	0.0423	−0.8921 ***
lnSTI	0.0208	0.0118	0.0059	0.0008	0.0103	−0.0155
Spatial						
ρ	−0.1490	−0.3893 ***	−0.1325	−0.1576	−0.3500 ***	0.3228 ***
Variance						
σ^2^ _e	0.0038 ***	0.0034 ***	0.0035 ***	0.0059 ***	0.0036 ***	0.0038 ***
Time effects	Yes	Yes	Yes	No	Yes	Yes
Individual effects	Yes	Yes	Yes	Yes	Yes	Yes
R^2^	0.8438	0.8037	0.7723	0.9431	0.9557	0.8183
AIC	−403.8642	−289.4422	−412.9957	−333.3009	−284.2851	−407.0186
BIC	−335.5334	−228.1174	−344.6649	−264.9701	−222.9603	−338.6878
N	165	120	165	165	120	165

Note: *, **, *** denote significant levels of 10%, 5%, and 1% levels, respectively.

**Table 12 ijerph-19-12441-t012:** Decomposition of sample effects of carbon emission quantity of urban production and energy consumption by region.

Variable	Eastern Region			Central Region			Western Region		
	Direct	Indirect	Total	Direct	Indirect	Total	Direct	Indirect	Total
lnIS	0.3009 *	−0.2644	0.0365	0.1678	0.3572 **	0.5251 *	−0.2950 **	−1.0332 ***	−1.3283 ***
lnFDI	−0.0123	0.0061	−0.0062	−0.0610 **	0.0329	−0.0281	0.0270 **	−0.0987 ***	−0.0717 ***
lnPCDI	1.1831 ***	−0.5774 ***	0.6057 *	1.4165 **	−0.5817	0.8348	0.0847	−1.1614 **	−1.0767
lnECS	0.0650 **	0.0221	0.0871	0.2040 ***	−0.0047	0.1993	0.4545 ***	−0.3217 *	0.1328
lnDREP	0.1187	−0.1478	−0.0290	−0.2641 *	0.0266	−0.2375	−0.1109	0.1399	0.0290
lnPD	0.0208	0.0825	0.1034	0.0119	0.1040 ***	0.1159 ***	−0.0247	−0.0064	−0.0311
lnGCR	0.1122	0.0145	0.1267	0.6026 ***	0.1598	0.7625*	0.1190	−0.1442	−0.0252
lnUR	0.3220	0.0129	0.3349	−0.1428	0.5972	0.4544	−2.3118 ***	−0.3062	−2.6180 ***
lnCL	0.1779	0.4766	0.6544	−0.4169	1.0451 *	0.6282	−0.5925 **	−0.6775	−1.2700 *
lnSTI	0.0189	0.0307	0.0496	0.0042	0.0365	0.0408	0.0081	−0.0366*	−0.0286

Note: *, **, *** denote significant levels of 10%, 5%, and 1% levels, respectively.

**Table 13 ijerph-19-12441-t013:** Decomposition of sample effects of carbon emission intensity of urban production and energy consumption by region.

Variable	Eastern Region			Central Region			Western Region		
	Direct	Indirect	Total	Direct	Indirect	Total	Direct	Indirect	Total
lnIS	0.4460 ***	−0.5545 *	−0.1085	−0.8449 ***	0.2351	−0.6098 **	−0.6809 ***	−1.8207 ***	−2.5016 ***
lnFDI	0.0203	0.0694 **	0.0896 **	−0.0382	0.0240	−0.0142	0.0049	−0.1382 ***	−0.1333 ***
lnPCDI	−0.5068 ***	−0.5983 ***	−1.1051 ***	0.4190	−2.1953 **	−1.7763 **	−0.4827	-0.9098	−1.3925 *
lnECS	0.0808 **	-0.0713	0.0095	0.2943 ***	0.0636	0.3579 ***	0.4692 ***	0.3759 **	0.8451 ***
lnDREP	0.2041 ***	-0.1237	0.0804	−0.1749	0.1281	−0.0468	−0.0403	0.5940 **	0.5537 **
lnPD	-0.0640	0.0172	−0.0468	−0.0177	0.0342	0.0165	−0.0248	−0.0138	−0.0386
lnGCR	-0.1184	−0.0068	−0.1251	0.9422 ***	0.2390	1.1812 ***	0.0394	−0.3587	−0.3194
lnUR	0.2040	1.5869 ***	1.7909 ***	0.0191	0.3844	0.4035	−2.3682 ***	−0.1752	−2.5434 ***
lnCL	-0.2411	0.4186	0.1775	−0.2053	1.3660 **	1.1607 *	−0.8434 ***	−1.3590 *	−2.2025 **
lnSTI	0.0004	−0.0110	−0.0105	0.0069	0.0207	0.0275	−0.0136	−0.0628 ***	−0.0764 **

Note: *, **, *** denote significant levels of 10%, 5%, and 1% levels, respectively.

**Table 14 ijerph-19-12441-t014:** Robustness test results of carbon emission quantity and carbon emission intensity.

Variable	lnCEQ	Direct Effect	Indirect Effect	Total Effect	lnCEI	Direct Effect	Indirect Effect	Total Effect
lnIS	0.2913 ***	0.2943 ***	0.4847 **	0.7790 ***	−0.2409 ***	−0.1775 **	1.0079 ***	0.8303 ***
lnFDI	−0.0363 ***	−0.0372 ***	−0.1390 ***	−0.1762 ***	−0.0405 ***	−0.0520 ***	−0.1921 ***	−0.2441 ***
lnPCCE	0.4643 ***	0.4685 ***	−0.0916	0.3769 *	−0.4942 ***	−0.4629 ***	0.4417 ***	−0.0212
lnECS	0.1015 ***	0.1009 ***	−0.0518	0.0491	0.1587 ***	0.1576 ***	−0.0200	0.1376
lnDREP	−0.0166	−0.0149	0.1772	0.1623	−0.0034	0.0167	0.3074 **	0.3240 **
lnPD	−0.0007	−0.0001	−0.0260	−0.0261	0.0073	0.0038	−0.0675	−0.0637
lnGCR	0.1540	0.1495	−0.2871	−0.1376	0.1893 **	0.1764 *	−0.2016	−0.0253
lnUR	0.4843 ***	0.4796 ***	0.0834	0.5630 *	0.3152 **	0.2570 *	−0.7837 **	−0.5267
lnCL	0.1050	0.1469	1.0464 *	1.1932 *	−0.6132 **	−0.5274 **	1.0432 **	0.5158
lnSTI	−0.0487 ***	−0.0487 ***	−0.0295	−0.0782 **	−0.0609 ***	−0.0663 ***	−0.0894 ***	−0.1557 ***
Spatial								
ρ	0.0639				0.3438 ***			
Variance								
σ^2^ _e	0.0101 ***				0.0104 ***			
Time effects	Yes				No			
Individual effects	Yes				Yes			
R^2^	0.7793				0.8990			
N	450				450			

Note: *, **, *** denote significant levels of 10%, 5%, and 1% levels, respectively.

## Data Availability

The data used to support the findings of this study are available from the first author upon request (e-mail: fuliyuan0123@163.com).

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
