# Peer review of "Spatial and Temporal Distribution and the Driving Factors of Carbon Emissions from Urban Production Energy Consumption"

_ijerph, 2022, doi:10.3390/ijerph191912441_

Round 1
Reviewer 1 Report
The article investigates carbon emission quantity and carbon emission intensity of urban production energy consumption in China between 2005 and 2019. The topic is important because carbon emissions must be reduced rapidly in order to mitigate global change. China is the largest emitter in the world which makes the paper even more topical and interesting.
The authors should consider the following comments and suggestions before publishing the article:
1) "Urban production energy consumption" could be defined and introduced better in the Introduction. The CEI calculation include terms "secondary industry" and "third industry" but it is not clear what they mean.
2) The article refers to China Energy Statistical Year book but that document is not included in the reference list.
3) Extra space between tables, figures and text could improve the lay-out of the article. Moreover, some spaces are missing in the text (at least rows 99 and 127). Section 2.3 starts with a lowercase initial.
4) It would be interesting to know why the regions are divided in the presented way (Section 2.7). Why the most Eastern parts in the North are included in the Central area?
5) At least the results of decomposition analysis (Tables 9 and 11) could be presented graphically. I believe the results would be more illustrative. Figures 3 and 4 could be narrower.
6) The place of sections 4 and 5 could be switched and the current section 4 (Discussion) could be named as Conclusions. In my opinion, there should be the Conclusions section.
Reviewer 2 Report
The authors addressed the hot burning issue of the environmental deterioration in China but there are some additional observations to improve the quality of the paper.
In introduction, authors are highlighting the issue of carbon emission in industrial sector. It is much better to incorporate some numerical figures about volume of carbon emissions in China and their significance for environmental deterioration. Moreover, the theoretical relationship among discussed variables will enhance the significance of the study.
It is also important to highlight that why this study is important and how is it novel?
A separate section of literature review will enhance the understanding of the readers about the topic.
The authors selected the 30 provinces of China covering the time span of 2005-2019 for empirical analysis so there should be some rationale for selecting the sample and time period.
There are different methods to compute the carbon emission quantity, Carbon emission intensity, so authors are suggested to incorporate that why they preferred the used methods in computation of these variables.
I would like to suggest to insert a table having the details of the variables used in the study like their abbreviation, their definition, their source etc.
The authors used the per capital GDP, per capita disposable income, per capita consumption expenditures, as independent variables. These variables are highly correlated among each other so there is need to address this issue. Moreover, the calculated indicator for education also needs to be reconsidered as the number of primary school students and number of junior middle school students have not any effect of carbon emission probably.
The findings of the study should be discussed and compared with earlier literature which will show that how this study is different from earlier studies.
Reviewer 3 Report
This study measures the quantity and intensity of 10 carbon emissions in 30 provinces of China based on urban production energy consumption, which have certain significance. but it still have some part need improved : 1. in the introduction part , most literature is listed, there is no classified discussion. 2. The empirical part of the paper lacks robustness test。3. when setting the model ,it should cite the related paper , to show why you choose these variebles .
Round 2
Reviewer 1 Report
The modifications made by authors improved the quality of the article.
Reviewer 2 Report
The authors incorporated the all observations.